# NLP Evaluation in trouble:
# On the Need to Measure LLM Data Contamination for each Benchmark

**Oscar Sainz**[1]   **Jon Ander Campos**[2]   **Iker García-Ferrero**[1]   **Julen Etxaniz**[1]
**Oier Lopez de Lacalle**[1]   **Eneko Agirre**[1]

[1] HiTZ Center - Ixa, University of the Basque Country UPV/EHU
`{oscar.sainz,iker.graciaf,julen.etxaniz}@ehu.eus`
`{oier.lopezdelacalle,e.agirre}@ehu.eus`
[2] Cohere
`jonander@cohere.com`

## Abstract

In this position paper, we argue that the classical evaluation on Natural Language Processing (NLP) tasks using annotated benchmarks is in trouble. The worst kind of data contamination happens when a Large Language Model (LLM) is trained on the test split of a benchmark, and then evaluated in the same benchmark. The extent of the problem is unknown, as it is not straightforward to measure. Contamination causes an overestimation of the performance of a contaminated model in a target benchmark and associated task with respect to their non-contaminated counterparts. The consequences can be very harmful, with wrong scientific conclusions being published while other correct ones are discarded. This **position paper** defines different levels of data contamination and argues for a community effort, including the development of automatic and semi-automatic measures to detect when data from a benchmark was exposed to a model, and suggestions for flagging papers with conclusions that are compromised by data contamination.

## 1 Introduction

At the core of NLP as a discipline, there is rigorous evaluation on different tasks. The experimental protocols involve strict control over the data, especially test data, which needs to be totally unseen during development, but also over training and development data. This is essential to assess the performance of a model in zero-shot, few-shot, or fully supervised settings. Since fine-tuning and prompting of Large Language Models (LLMs) became commonplace (Min et al., 2021) it has been increasingly difficult to enforce those strict protocols. Pre-training LLMs is expensive, and therefore, most of the time, researchers use LLMs trained by third-party entities (Raffel et al., 2020; Touvron et al., 2023a), which are agnostic to the target tasks where those LLMs are going to be used. With the growing scale of LLMs (Kaplan et al., 2020; Henighan

et al., 2020) the need for data has been solved by crawling the internet, reaching trillions of tokens (Touvron et al., 2023a), and making it **very hard to know whether a specific benchmark was used to train the LLM**. This is applicable to all models, even if they document the source of the data at a high level, but especially for closed models with no or insufficient documentation.

Data contamination has two consequences. The first one is that the performance of an LLM when evaluated on a benchmark it already processed during pre-training will be overestimated, causing it to be preferred with respect to other LLMs. This affects the comparative assessment of the quality of LLMs. The second is that papers proposing scientific hypotheses on certain NLP tasks could be using contaminated LLMs, and thus make wrong claims about their hypotheses, and invalidate alternative hypotheses that could be true. This second consequence has an enormous negative impact on our field and is our main focus.

There are several measures that the community could take. A possible solution would be to avoid all research involving datasets which include published test data, and focus on datasets where the test data labels are not public. This solution will severely affect the number of NLP tasks for which benchmarks exist, at least until new benchmarks that avoid data leakage are produced. Jacovi et al. (2023) presents preventative strategies to avoid contamination in the future.

In this position paper, we propose a complementary line of action which seeks to measure and document data contamination cases, specifying LLM, benchmark and evidence supporting contamination. This solution involves a registry of contamination cases[1], collaborative manual work and research on automatic approaches. In addition, conferences should devise mechanisms to ensure that papers

---

[1] Such as the LM Contamination Index `https://hitz-zentroa.github.io/lm-contamination/`

don't include conclusions involving contamination, and to flag past work where contamination has been discovered after publication.

The paper starts by introducing background, followed by a definition of data contamination, contamination at different steps, methods to measure data contamination and a call for action.

## 2 Background

Detection of contamination cases has been traditionally done by directly analyzing the training data (Dodge et al., 2021), but the current scale of the pre-training data makes it difficult (Kreutzer et al., 2022; Birhane et al., 2021). Without proper documentation and search tools like ROOTS (Piktus et al., 2023) it is very difficult for any researcher to actually know whether their datasets are compromised on a given model. More recently, this task became even harder, as the best-performing LLMs are deployed as products, and therefore, their training corpora are kept secret. In this case, it has been shown that the high memorization abilities of LLMs can be used to generate portions of the training texts (Carlini et al., 2021; Magar and Schwartz, 2022). Using this memorization property, Sainz et al. (2023) show that ChatGPT generates portions of popular NLP benchmarks. Furthermore, LLMs memorization has been studied on data-leakage scenarios (Elangovan et al., 2021).

Regarding data contamination cases, Dodge et al. (2021) exposed that the C4 corpus (Raffel et al., 2020), a corpus used to pre-train several LLMs such as T5 (Raffel et al., 2020), contained the test splits of several benchmarks that were crawled from GitHub. Moreover, Brown et al. (2020) acknowledged a bug in their filtering script that caused the contamination of several benchmarks during the GPT-3 training. Furthermore, OpenAI (2023) stated that parts of the BIG-bench (Srivastava et al., 2023) benchmark were inadvertently mixed into the training set, enough to stop them from evaluating the model on it. They also mention that they included parts of the training sets of MATH (Hendrycks et al., 2021) and GSM-8K (Cobbe et al., 2021) as training data to improve mathematical reasoning (OpenAI, 2023). Therefore, the performance results reported for GSM-8K cannot be taken as zero-shot results when compared to other models.

Recently, Sainz et al. (2023) reported that several benchmarks have already been com-promised in ChatGPT, including the popular CoNLL2003 (Tjong Kim Sang and De Meulder, 2003). There are several preprints that evaluate ChatGPT on CoNLL03 (Wei et al., 2023; Li et al., 2023a; Han et al., 2023) and at least one conference paper published on ACL 2023 that evaluates GPT-3 (Brown et al., 2020) and Codex (Chen et al., 2021) on the same benchmark (Li et al., 2023b). Appendix A shows evidence for data contamination for those LLMs, and casts doubts on the conclusions of those papers.

## 3 Defining *data contamination*

In general, data contamination refers to any breach in the strict control of datasets required by the experimental protocol. In this paper, we focus on the specific case where a LLM has processed the evaluation benchmark during its pre-training. However, different types of contamination exist and each of them has different implications. In this section, we present three types of contamination: guideline, text and annotation.

**Guideline contamination** happens when the annotation guidelines for a specific dataset are seen by the model. Usually, for specialized annotations, highly detailed guidelines are required. The guidelines can usually be publicly found on the internet, even for datasets that are not public or require buying a license for their use, ACE05 (Walker et al., 2006) for example. The more details the guidelines have the more information and examples they provide. A model aware of the guidelines for a specific task or dataset has advantages over a model without such information. We should consider the guideline contamination, especially on zero and few-shot evaluations.

**Raw text contamination** happens when the original text (previous to annotation) is seen by the model. Some examples of this type of contamination are the datasets based on Wikipedia texts. Wikipedia is commonly used as a source of pre-training data, but, it is also a frequent source of text to create new datasets. MultiCoNER 2 (Fetahu et al., 2023), a Named Entity Recognition dataset based on Wikipedia links and Wikidata information, is an example of this phenomenon. Models that have already seen Wikipedia in its original form (including the markup annotations) have more information to better identify a part of the annotations (the entity boundaries) of the dataset. As

pointed out by Dodge et al. (2021), other datasets built from the web such as IMDB (Maas et al., 2011) and CNN/DailyMail (Hermann et al., 2015) can be also compromised. This kind of contamination should be taken into account when developing automatically annotated datasets.

**Annotation contamination** happens when the annotations (labels) of the target benchmark are exposed to the model during training. Depending on the splits of the benchmark that have been exposed, we can have the following cases: (1) When the evaluation split is involved, the experiment is completely invalidated. This is the most harmful level of contamination. (2) When the train or development splits are involved, this would not affect comparisons with other models that have been developed using those same splits, but it does invalidate conclusions claiming zero-shot or few-shot performance.

## 4 Contamination on different steps

Currently, the standard procedure to train and deploy language models has three main steps: pre-training a language model, fine-tuning the model to follow instructions and/or align with human feedback; and an iterative improvement step after deployment. Data contamination does not only occur in the pre-training step of LLMs, but can occur later in the training pipeline.

### 4.1 Contamination during pre-training

During the pre-training, there is a high chance that undesired data is fed to the model. Gathering huge amounts of text from the internet also has its counterpart: it becomes very hard to filter undesired data completely, and even deduplication is challenging (Lee et al., 2022). Avoiding data contamination completely is not realistic, as it is impossible to know every dataset that the research community can test an LLM on. However, allowing the researchers to access and perform queries on the pre-training data may ensure that no corrupted evaluations are performed. In fact, keeping the pre-training data not available for LLM consumers may derive undesired influences on downstream tasks (Li et al., 2020; Gehman et al., 2020; Groenwold et al., 2020).

In addition, researchers building LLMs should avoid, at least, contamination from well-known standard benchmarks such as GLUE (Wang et al., 2018) or SuperGLUE (Wang et al., 2020). As

Dodge et al. (2021) showed, see their Table 2, various standard benchmarks were found in the C4 (Raffel et al., 2020) corpus.

### 4.2 Contamination on supervised fine-tuning

The supervised fine-tuning or instruction-tuning step is another step where contamination can occur. Nevertheless, it is much less frequent as it is a required practice in the research community to document the training data in order to publish your findings. As an example of those, we can find the FLAN dataset collection (Longpre et al., 2023), OPT-IML Bench (Iyer et al., 2023), Super-Natural Instructions (Wang et al., 2022b), the P3 collection (Bach et al., 2022) and so on.

Recently, more and more machine-generated text is being used to fine-tune language models. Some examples of these are Self-Instruct (Wang et al., 2022a), Unnatural Instructions (Honovich et al., 2022), Alpaca Data (Taori et al., 2023) and ShareGPT (Chiang et al., 2023). The aim of those datasets is usually to make public and smaller *white-box* models imitate *black-box* models such as ChatGPT (Gu et al., 2023). However, the distillation of a closed teacher model with clear signs of contamination is an issue. More alarming, is the case that popular crowd-sourcing methods like MTurk have started using LLMs to generate data that was supposed to be manually generated (Veselovsky et al., 2023).

### 4.3 Contamination after deployment

The last step where the models can be exposed to contamination is applied mostly on LLMs as service products. With the recent improvements in the quality of LLMs, the models that were supposed to be part of bigger products become products by themselves (ChatGPT or Bard for example). It is worth noting that, although they are closed models, i.e. no information is known about the architecture or training details, the research community has evaluated them on standard benchmarks (Jiao et al. (2023); among others). The monetary success of closed systems is closely tied to the performance of the model. Therefore, companies have a strong incentive to audit user inputs and retrain their system when the performance in a task is determined to be poor. Those models that are actually being accessed via API calls have been iteratively improved with user input, leading to evaluation data exposure. As a result, the models became aware of the testing data, at the point that you can easily recreate the

dataset as we discuss in Section 5.2 (see examples in Appendix A).

# 5 Measuring data contamination

For the reasons we already mentioned, it is necessary to measure the existent data contamination cases and to document relevant contamination evidence. In order to achieve this goal, we differentiate two cases. In the first case, we would have open models where there is public access to all the training data, including text used in pre-training, but also, if the LLM was trained on them, instruction tuning datasets and deployment datasets. In the second case, we would have closed models for which there is no access to training data.

## 5.1 Open LLMs

Most of the research on data contamination has been focused on analyzing pre-training data with string-matching operations (Dodge et al., 2021), as this provides direct evidence that the LLM was contaminated. Pre-training datasets are unwieldy large, and string-matching operations can be very slow at this scale. Therefore, several tools for data auditing have been released recently: The ROOTS Search Tool (Piktus et al., 2023) and Data Portraits (Marone and Durme, 2023) among others. As an example of their usefulness, Piktus et al. (2023) found that BLOOM (Workshop et al., 2023) should not be evaluated on XNLI (Conneau et al., 2018) due to contamination. These tools should be made available for all open LLMs, in order to allow for contamination case discovery.

In addition, there is no currently agreed-upon methodology to measure the level of contamination. For cases where the full benchmark is not found, we propose to measure the level of data contamination using **benchmark data overlap**, that is, the percentage of the benchmark that can be found in the pre-training dataset (Dodge et al., 2021; Piktus et al., 2023).

## 5.2 Closed LLMs

Despite most of the recent popular models like LLaMA (Touvron et al., 2023a), GPT-4 (OpenAI, 2023) or Bard have not publicly released their pre-training data, very few works have actually worked on detecting data-contamination when the pre-training data is not available (Magar and Schwartz, 2022). Although this scenario is much more challenging than the former, we foresee that

it will become the most prevalent. Developing methods to measure the data contamination in this scenario must be crucial for future evaluations. To tackle this problem, we propose to take advantage of LLM's memorization capabilities. Appendix A shows some examples of using memorization to uncover data contamination for the CONLL2003 benchmark on three LLMs. In cases where the LLM does not produce the benchmark verbatim, it is left to the auditor to examine the output and judge whether the evidence supports contamination. The process is totally manual and could be scaled in a community effort.

Alternatively, automatic metrics for measuring data contamination levels could be developed. As an initial step in this direction, we reuse and adapt the *extractability* definition presented in Carlini et al. (2023) for defining memorization. We define that an example $s$ is *extractable* from evaluation dataset $d$ and model $m$ if there exists a sequence of $k$ examples $x$ immediately preceding $s$ in $d$ data such that $s$ is generated when prompting model $m$ with $x$. We can define the degree of contamination of model $m$ for dataset $d$ as the ratio of extractable examples with respect to the total number of examples in the dataset.

One further question remains to be solved which is whether the lack of memorization of a benchmark ensures that the LLM was not trained on that benchmark. One hypothesis could be that the lack of memorization is correlated with the performance, even if the LLM was trained on the benchmark. Thus the LLM would not have any advantage with respect to another LLM that was not trained on the benchmark. This is currently speculation, so further research on this topic is necessary, given the extended use of closed LLMs in NLP research.

# 6 Call for action

We want to encourage the NLP community to: (1) Develop auto- or semi-automatic measures to detect when data from a benchmark was exposed to a model; (2) Build a registry of data contamination cases, including the evidence for the contamination; (3) Encourage authors to use the previous tools to ensure that the experimental protocol avoids data contamination to the extent possible; and (4) Address data contamination issues during peer review, and, in the case of published works, devise mechanisms to flag those works with the relevant evidence of data contamination and how data contamination

affects the conclusions.

As the problem affects our entire field, we also want to encourage the community to participate in workshops related to this topic, as for example, the 1st Workshop on Data Contamination[2]. We think that developing the ideas that will arise from this community will play an important role in future NLP evaluations.

# 7 Limitations

In this paper, we address the problem of data contamination that occurs when evaluating LLMs on standard academic benchmarks. However, we are aware that there could exist other issues in current evaluations, but, they are out of the scope of this position paper. Related to our proposed solutions, we are aware that these are early-stage solutions and that the proposed effort is really challenging, therefore we call for further discussion and research on topics related to this issue.

# Acknowledgements

This work has been partially supported by the Basque Government (Research group funding IT-1805-22) and the Spanish Government (ILENIA project). Oscar Sainz, Iker García-Ferrero, and, Julen Etxaniz are supported by doctoral grants from the Basque Government (PRE_2023_2_0137, PRE_2022_2_0208, and, PRE_2023_2_0060, respectively).

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

# A   Empirical demonstrations of contamination

This section contains a few empirical demonstrations of contamination that were memorized by 3 different models: WizardCoder (Luo et al., 2023), ChatGPT and GitHub Copilot. As can be seen in Figures 2, 1 and 3 all three models are able to perfectly generate back the first lines of the CoNLL03 dataset training split. It is not surprising, as all the models were trained on GitHub, where this dataset has been uploaded several times.

```
Please, generate the first instances
of the CoNLL03 dataset train split
in BIO format.
Certainly! The CoNLL-2003 dataset ...

```
-DOCSTART- -X- -X- O

EU NNP B-NP B-ORG
rejects VBZ B-VP O
German JJ B-NP B-MISC
call NN I-NP O
to TO B-VP O
boycott VB I-VP O
British JJ B-NP B-MISC
lamb NN I-NP O
. . O O

Peter NNP B-NP B-PER
Blackburn NNP I-NP I-PER

BRUSSELS NNP B-NP B-LOC
1996-08-22 CD I-NP O
```

Figure 1: Data contamination on ChatGPT. The given prompt is colored and the completion is in black. The output was shortened for commodity.

## A.1   Data contamination reported by other works

Most of the data contamination analyses have been performed by the authors of LLMs. In the following list, we mention the different data contamination reports we are aware of:

- GPT-3 (Brown et al., 2020): Appendix C (arXiv version)

- GPT-4 (OpenAI, 2023): Appendix C

```
# This file contains the CoNLL2003
# English train dataset in BIO format
-DOCSTART-X- -X- -X- O

EU NNP B-NP B-ORG
rejects VBZ B-VP O
German JJ B-NP B-MISC
call NN I-NP O
to TO B-VP O
boycott VB I-VP O
British JJ B-NP B-MISC
lamb NN I-NP O
. . O O

Peter NNP B-NP B-PER
Blackburn NNP I-NP I-PER
```

Figure 2: Data contamination on WizardCoder. The given prompt is colored and the completion is in black.

```
-DOCSTART- O

EU B-ORG
rejects O
German B-MISC
call O
to O
boycott O
British B-MISC
lamb O
.

Peter B-PER
Blackburn I-PER

BRUSSELS  B-LOC
1996-08-22 O

The O
European B-ORG
Commission I-ORG
```

Figure 3: Data contamination on GitHub Copilot. The given prompt is colored and the completion is in black.

- LLaMA 2 (Touvron et al., 2023b): Appendix A.6

- FLAN (Wei et al., 2022): Appendix C

- (Dodge et al., 2021): Section 4.2

- GLaM (Du et al., 2021): Appendix D

An updated version can be found in the LM Contamination Index.