# OpenReview forum: "NLP Evaluation in trouble: On the Need to Measure LLM Data Contamination for each Benchmark"
_EMNLP/2023/Conference — EMNLP 2023 Findings_

### Official Review · Reviewer_z6oP · 2023-07-23

**Soundness:** 3

**Excitement:**

3: Ambivalent: It has merits (e.g., it reports state-of-the-art results, the idea is nice), but there are key weaknesses (e.g., it describes incremental work), and it can significantly benefit from another round of revision. However, I won't object to accepting it if my co-reviewers champion it.

**Missing References:**


1. Memorization vs. Generalization : Quantifying Data Leakage in NLP Performance Evaluation (Elangovan et al., EACL 2021)

**Paper Topic And Main Contributions:**

This is a position paper that discusses how with LLMs, the training data may not be publicly disclosed / well documented creating challenges in NLP performance evaluation. It  conceptually offers "contamination report registry" as a way to crowdsource contamination as well as encourage authors to verify if the evaluation sets haven't already been consumed by the LLM

**Questions For The Authors:**

A). LIne 150 - When you refer to annotation guideline does this also include the  task? One of the prominent aspect of extra large LLMs is their ability to following natural language instructions from a everyday person. So have you considered the impact of how these LLMS are impacted if they have been trained specifically for a given task that can be described in many different ways?

**Reasons To Accept:**

- Evaluation using test sets is one of  the core foundations of NLP and this paper highlights  that with LLMs this approach of simply using a test set needs to be revisited given the LLMs might have already consumed the test data. This paper highlights a very important topic and that is the main reason to accept.

**Reasons To Reject:**

- While the call for use of a "contamination registry" as a possible way of verifying contamination, this paper doesn't not provide details of how such a registry would be designed and what the practical constraints of such a registry would be.

- While the paper mainly focusses on contamination, even if we address contamination the broader question remains *"Are the current test sets even effective to detect the weakness / strengths of a powerful model such as GPT3-175B ?"*.  LLMs with over billion parameters have become extremely powerful, so a question that is not addressed in this paper is the effectiveness of standard test sets, given that existing NLP public test sets evaluate limited capabilities/ weakness of even much smaller pretrained models such as BERT that have a few hundred million parameters.

- Generative tasks ( given popular models  such as GPT, Llama are generative) are generally difficult to automatically evaluate given standard metrics such as Rouge etc are inadequate in reflecting human evaluation. The paper doesn't explicitly cover the challenge of evaluating generative tasks especially in the content of extremely fluent LLMs such as GPT as most papers in NLP report automated metrics as human evaluation tends to be expensive,  time consuming and difficult to replicate.

The main objective of my comments is not to increase the scope of the paper, but rather  critically think about *"Would a contamination registry be really useful in evaluating LLMs? If not, what would problems would overshadow/ diminish  the relevance of a contamination registry"*


**Reproducibility:**

N/A: Doesn't apply, since the paper does not include empirical results.

**Reviewer Confidence:**

3: Pretty sure, but there's a chance I missed something. Although I have a good feel for this area in general, I did not carefully check the paper's details, e.g., the math, experimental design, or novelty.

---

> ### Author Rebuttal · Authors · 2023-08-26
>
> Thanks for your valuable comments, which we found most helpful for improving the overall quality of our paper.
>
> **About providing details of how a "contamination registry" would be designed:**
>
> An example of contamination registry as a possible solution could be the one proposed by Sainz et al., (2023) (cited in the paper) that can be found at https://hitz-zentroa.github.io/lm-contamination/ . However, many improvements are needed in order to make it practically viable:
> * A unified and standard methodology to measure contamination.
> * Open and documented pre-training datasets in order to easily test the contamination on new benchmarks.
> * [In case the previous is not possible] The development of a measure that estimates the level of contamination for any given pair of models and benchmark.
> * Information about the potential impact on the evaluations.
>
> The practical constraints of a contamination registry are, in fact, the requirements mentioned above. Once the requirements are satisfied, the registry is a database that researchers can consult in order to decide with which model and dataset pairs perform the evaluations.
>
> If the paper is accepted we will add a section giving more details about it.
>
> **About "Are the current test sets even effective to detect the weakness / strengths of a powerful model such as GPT3-175B?**
>
> The effectiveness of the current evaluation benchmarks is not covered in this paper, as this is a complementary issue to that of data contamination. Although there is discussion on the effectiveness of current test sets, papers reporting results and leaderboards are currently active, and NLP practitioners do take conclusions based on those paper results and leaderboards. Our paper focuses on whether the reported results are correct and comparable, which is still a relevant problem. In other words, while the community continues to use current test sets, our work is relevant. We will mention this explicitly in the paper.
>
> Note that the problem of data contamination does not only apply to current benchmarks, but future benchmarks as well. However, there are a lot of benchmarks that remain challenging even for “powerful models such as GPT3-175B”, to give you an example, GPT3-175B performs as good as a random baseline on benchmarks like ANLI or WiC. Even if some of the benchmarks become obsolete, there will be new ones, and the problem of data contamination will remain unsolved.
>
> **About generative tasks:**
>
> We agree that evaluating generative tasks is challenging. However, that isn’t the main focus of our paper. In fact, data contamination is one of the challenges when evaluating models on both discriminative and generative tasks.
>
> **About _"Would a cotamination registry be really useful in evaluating LLMs?"_**
>
> Although the evaluation of generative LLMs is an open problem, the current practice of reporting results on test sets is still one of the main selling points of such models. And it’s not only an issue of evaluating generative LLMs. Note also that “traditional” NLP tasks that are evaluated with test sets are the focus of countless papers in our conferences. If those papers use generative LLMs which are contaminated for those test sets, the conclusions become incorrect.
>
> Unless action is taken the problem of data contamination will persist, and no one should trust contaminated evaluation results. A contamination registry will give researchers the information about which benchmarks/models are contaminated on a given model and which not, to properly draw their conclusions. In parallel, as we mention in the paper, other suggestions like preventing contamination in future datasets are also necessary (Jacovi et al., 2023).
>
> **Question A: When you refer to annotation guideline dows this also include the task? ...**
>
> We understand that by “this also includes the task” the reviewer refers to the description of the task. Yes, usually the annotation guidelines do include the general description of the task, in addition to detailed definitions of labels and examples of how the annotators should proceed. It is well known that LLMs transfer knowledge easily between tasks. From the point of view of data-contamination, as long as the use of guidelines to train the LLM is properly reported, this can be understood as regular transfer learning, and is not an instance of data contamination.
>
> **About missing references:**
>
> Thanks for the missing reference. We will include it in our final version.

---

### Official Review · Reviewer_stGG · 2023-08-04

**Typos Grammar Style And Presentation Improvements:** None
**Soundness:** 3

**Excitement:**

3: Ambivalent: It has merits (e.g., it reports state-of-the-art results, the idea is nice), but there are key weaknesses (e.g., it describes incremental work), and it can significantly benefit from another round of revision. However, I won't object to accepting it if my co-reviewers champion it.

**Missing References:**

None

**Paper Topic And Main Contributions:**

The paper investigated the data contamination in the NLP domain and provided a definition of different levels of data contamination. Further, the authors gave several suggestions on the development of measures of the detection of data contamination.

**Questions For The Authors:**

(1) Is the term "data contamination" first formally proposed in this paper?

(2) Can the authors provide some empirical statistics and examples to show the importance of the data contamination to the NLP models?

**Reasons To Accept:**

A remarkable advantage is that the paper provides a comprehensive investigation of data contamination from different perspectives for NLP practitioners. The investigation identifies key directions for further research on NLP data contamination.

**Reasons To Reject:**

The authors should provide some empirical results,  figures, and examples in the study of data contamination in order to make the paper more attractive.

**Reproducibility:**

N/A: Doesn't apply, since the paper does not include empirical results.

**Reviewer Confidence:**

4: Quite sure. I tried to check the important points carefully. It's unlikely, though conceivable, that I missed something that should affect my ratings.

---

> ### Author Rebuttal · Authors · 2023-08-26
>
> Thank you for your effort and insightful comments.
>
> **About empirical results, figures, and examples:**
>
> The state of data contamination research is premature, with several open questions. In order to perform empirical evaluations it would be necessary to have a standardized methodology to measure the extent of data contamination, but as we pointed out in the paper, this  is not straightforward. In fact, several influential papers have included sections on data contamination, but the methods used to measure contamination are completely different.
>
> This is a position paper that aims to make the community aware of the data contamination problem and give some definitions and directions on how to address the problem. The paper already cites a few studies of data contamination (Dodge et al., 2021; Carlini et al., 2021; Piktus et al., 2023; among others) and also provides some examples in the Appendix A.
>
> However, we agree that some empirical results would be helpful to understand the severity of the problem. We will extend Appendix A to include some of the results reported on the papers we cited, in addition to additional examples.
>
> **Question 1: Is the term "data contamination" first formally proposed in this paper?**
>
> The term of data contamination was introduced by Brown et al., (2020). Later works have also made use of the term, but there is no formal and agreed upon definition. In this paper we propose such a definition.
>
> **Question 2: Can the authors provide some empirical statistics and examples to show the importance of the data contamination to the NLP models?**
>
> We will include the following results in the Appendix for the camera ready version:
> * Brown et al., 2020 (ArXiv version - https://arxiv.org/abs/2005.14165) Appendix C
> * OpenAI, 2023 (https://arxiv.org/abs/2303.08774) Appendix C
> * Touvron et al., 2023 (https://arxiv.org/abs/2307.09288) Appendix A.6
> * Wei et al., 2022 (https://openreview.net/forum?id=gEZrGCozdqR) Appendix C
> * Dodge et al., 2021 (https://aclanthology.org/2021.emnlp-main.98/) Section 4.2
> * Du et al., 2022 (https://arxiv.org/abs/2112.06905) Appendix D

---

### Official Review · Reviewer_mDTS · 2023-08-04

**Soundness:** 3

**Excitement:**

3: Ambivalent: It has merits (e.g., it reports state-of-the-art results, the idea is nice), but there are key weaknesses (e.g., it describes incremental work), and it can significantly benefit from another round of revision. However, I won't object to accepting it if my co-reviewers champion it.

**Paper Topic And Main Contributions:**

This position paper highlights the problem of data contamination in NLP evaluations, particularly when large language models (LLMs) are trained on the test split of a benchmark and then evaluated on the same benchmark. The authors argue that this contamination leads to an overestimation of model performance and can result in incorrect scientific conclusions being published. They propose a community effort to develop measures for detecting data contamination, a registry to document contamination cases, and mechanisms to flag papers with compromised conclusions. The paper emphasizes the need for transparent evaluation protocols and suggests tools to aid in detecting contamination.

**Reasons To Accept:**

This paper identifies data contamination as a critical issue in NLP evaluations when LLMs are trained on benchmark test data and then evaluated on the same benchmarks.

The authors propose the following practical solutions: detecting contamination, building a registry, and promoting transparent evaluation protocols.

The benefits to the NLP community: improved evaluation practices, increased awareness, reduced risk of misleading conclusions, consistent evaluation standards, and advancements in benchmarking.

**Reasons To Reject:**

- Lack of Empirical Evaluation: The paper lacks empirical evaluations or case studies to demonstrate the extent of data contamination in real-world scenarios.

- The paper primarily focuses on data contamination caused by training LLMs on benchmark test data. It may not address other potential sources of bias or contamination in NLP evaluations, which could limit its overall impact on the field.

- Overemphasis on Closed Models: The paper heavily discusses closed LLMs, which might not be representative of all NLP research.

- Ambiguity in Measuring Contamination: The paper proposes measuring data contamination but lacks a clear, standardized methodology. The proposed method using string-matching operations may not be sufficient for large-scale evaluations, and there is a need for more robust automatic measures.

**Reproducibility:**

N/A: Doesn't apply, since the paper does not include empirical results.

**Reviewer Confidence:**

3: Pretty sure, but there's a chance I missed something. Although I have a good feel for this area in general, I did not carefully check the paper's details, e.g., the math, experimental design, or novelty.

---

> ### Author Rebuttal · Authors · 2023-08-26
>
> Thank you very much for your thorough review, we appreciate your comments that we believe will improve the paper.
>
> **About Lack of Empirical Evaluation and Ambiguity in Measuring Contamination:**
>
> The state of data contamination research is premature, with several open questions. In order to perform empirical evaluation it would be necessary to have a standardized methodology to measure the extent of data contamination, but as we pointed out in the paper, this  is not straightforward. In fact, several influential papers have included sections on data contamination, but the method to measure contamination is completely different.
>
> The extent of data contamination is also unknown. We gathered papers and works on data contamination and examples (Appendix A) which show that data contamination is not a marginal problem. It is currently affecting the conclusions in papers accepted in top conferences, and unless the community takes action, might have serious negative effects in the field.
>
> Given this premature state, the goal of this position paper is not to define a methodology but to engage researchers to work towards that direction. That is, our proposed solutions are initial ideas. As we say in the paper “there is no currently agreed-upon methodology to measure the level of contamination” this is a topic that needs to be discussed in the community.
>
> If we had a methodology which we thought would be effective and solve the issue, we wouldn't have sent a position paper, but an empirical paper. We will clarify the issue about the current premature status in the paper.
>
> **About the focus on data contamination caused by training LLMs on benchmark test data:**
>
> Bias of LLMs is a very relevant problem which has been the focus of several papers and workshops. We view data contamination as a very different, complementary, problem which deserves a separate focus. In the same sense there are other issues with empirical evaluation which are also complementary. We will mention this complementarity explicitly in the paper.
>
> **About Overemphasis on Closed Models:**
>
> The paper does cover both in Section 5. There are many more closed models and the most popular models are closed. Note that models like LlaMA are considered closed for the sake of data contamination: although the model is openly available, the pre-training data used is not. In addition, measuring data contamination in closed models is more difficult, and methods that apply to closed models also apply to open models. The current writing does not mention this explicitly, so we will clarify this in the final version.

---

### Meta-Review · Area_Chair_w3uh · 2023-09-24

**Recommendation:** 4

**Metareview:**

This short position paper stresses the need of revisiting our standard practice of evaluating and benchmarking our progress using publicly available test sets, due to data contamination issues in large pretrained models. The paper argues for a need for community effort in identifying and flagging data contamination in available benchmarks.

Pros:
* This is a very timely paper discussing data contamination as a critical issue in NLP evaluations. The paper makes several suggestions and identifies important directions for further research in this area.
* The authors proposed a community wide effort in detecting data contamination, building a contamination registry, and promoting transparent evaluation protocols.

Cons:
* The reviewers have pointed out that the paper lacks any empirical results or a clear standardized methodology to identify and measure data contamination.

While I agree with the reviewers that the paper would improve (and be more impactful) with empirical evidence or a concrete plan for the contamination registry, I agree with the authors’ rebuttal that we are in a dire need of a wide discussion in our community on this critical issue. As a short position paper, it will raise awareness and engage practitioners to work towards that direction.

Other concerns were adequately addressed in the rebuttals.  If accepted, please include discussion addressing all the raised concerns to improve the paper.

---

### Decision · Program_Chairs · 2023-10-07

**Decision:**

Accept-Findings

**Comment:**

This short position paper stresses the need of revisiting our standard practice of evaluating and benchmarking our progress using publicly available test sets, due to data contamination issues in large pretrained models. The paper argues for a need for community effort in identifying and flagging data contamination in available benchmarks.

Pros:
* This is a very timely paper discussing data contamination as a critical issue in NLP evaluations. The paper makes several suggestions and identifies important directions for further research in this area.
* The authors proposed a community wide effort in detecting data contamination, building a contamination registry, and promoting transparent evaluation protocols.

Cons:
* The reviewers have pointed out that the paper lacks any empirical results or a clear standardized methodology to identify and measure data contamination.

While I agree with the reviewers that the paper would improve (and be more impactful) with empirical evidence or a concrete plan for the contamination registry, I agree with the authors’ rebuttal that we are in a dire need of a wide discussion in our community on this critical issue. As a short position paper, it will raise awareness and engage practitioners to work towards that direction.

Other concerns were adequately addressed in the rebuttals.  If accepted, please include discussion addressing all the raised concerns to improve the paper.